# Evaluation of an Innovative Zn Source on Feed Efficiency, Growth Performance, Skin and Bone Quality of Broilers Suffering Heat Stress

**DOI:** 10.3390/ani12233272

**Published:** 2022-11-24

**Authors:** Mojtaba Zaghari, Hossein Mehrvarz, Hosna Hajati, Hossein Moravej

**Affiliations:** 1Department of Animal Science, College of Agriculture and Natural Resources, University of Tehran, Karaj 1417466191, Iran; 2Animal Science Research Department, East Azarbaijan Agricultural and Natural Resources Research and Education Center, AREEO, Tabriz 5153715898, Iran

**Keywords:** broiler, performance, heat stress, HiZox, zinc oxide

## Abstract

**Simple Summary:**

Heat stress is detrimental for birds, and it can affect a bird’s performance negatively. Zinc is essential in enzymes critical for the integrity of the cells involved in heat stress. Thus, it seems that using a desired source of zinc can improve the health status of birds.

**Abstract:**

One thousand two hundred male broilers were used to evaluate the effect of different dosages of HiZox^®^ on feed efficiency, growth performance and bone quality of broilers suffering from heat stress. A completely randomized design was used, with four treatments and ten replicates. Basal corn–soybean meal diets supplemented with 75, 100 and 125 mg/kg zinc from HiZox and 100 mg/kg zinc from regular ZnO were used to make four treatments. Heat stress was induced after the third week by keeping house temperature between 28–34 °C, from 1 pm until 5 pm. The body weights of the birds that received the diet supplemented with HiZox or ZnO showed no significant difference at 7 and 14 days. Body weight of heat stressed birds fed diets containing different levels of HiZox or ZnO were not different at 28 and 42 days of age. In comparison to the Ross 308 management guide, induced heat stress diminished body weight and feed intake by approximately 17 and 21%, respectively. At 28 days, chickens who received 125 mg/kg Zn from Hizox had better feed efficiency (*p* < 0.05). The mortality rate of heat-stressed male broiler chickens who received different dosages of HiZox was 2.85% less than that of the regular ZnO group (*p* < 0.06). The results showed that addition of HiZox to the diet of male broiler under heat stress doubled the skin resistance during feather plucking in the slaughter plant and improved carcass quality (*p* < 0.07). Tibia breaking strength, included elongation and extension were improved by consumption of a diet supplemented with 75 mg HiZox/kg (*p* < 0.09). The HiZox-75 fed broilers required higher amounts of energy (MJ) for tibia breaking at break and peak points at 42 days (*p* < 0.09; *p* < 0.07). Jejunum Zn concentrations reflected the quantity of ingested Zn (*p* < 0.0001). Gizzard Zn solubility was dependent on dietary treatment (*p* < 0.03). Solubility of Zn in the gizzard of chickens who received HiZox was higher (about 30%) than broilers fed regular ZnO. In conclusion, Zn from HiZox was more efficient in decreasing heat stress mortality, increasing skin resistance and bone breaking strength compared to a regular ZnO source.

## 1. Introduction

Heat stress, characterized by the imbalance of the poultry house temperature and body heat dissipation, has a significant negative influence on poultry performance and the income of farms around the world [1]. Due to genetic improvements and producing broilers with higher metabolic rates, nowadays, broiler chickens are more susceptible to heat stress. High ambient temperatures can suppress immune system, increase mortality and decrease feed efficiency and growth performance of the birds [2]. Zinc is one of the critical minerals that can increase the effectiveness of vitamin E during heat stress conditions [3]. It is well known that zinc must be present in the diets of all animals, and must be supplied almost continuously, because animals have only small amounts of readily available stored Zn in their bodies [4]. In fact, natural Zn concentrations in feedstuffs are generally lower than the daily Zn requirement for broilers, leading to the necessity of dietary Zn supplementation. Zinc participates as a cofactor or component in more than 300 enzymes, being important for protein and carbohydrate metabolism, growth and reproduction. It has been found that there is a wide range, from 10 to 110 mg kg^−1^, in broiler diets for zinc requirement [5]. The NRC [6] recommends a level of 40 to 75 mg/kg of Zn in various poultry diets; however, commercial broiler companies have announced higher amounts, such as a 100 mg Zn/kg diet (Cobb 500, 2018), or a 100 mg Zn/kg diet (Ross 308, [7]). In addition, research has shown that heat stress induces more excretion of zinc. When considering Zn as a key cofactor for antioxidant enzymes, such as superoxide-dismutase, the importance of the issue magnifies [8]. It has been reported that ZnO at high doses is an efficient treatment for prophylaxis diarrhea in piglets [9,10]. Interestingly, supplemental Zn has a positive effect on tibia development and intestinal microflora population in broilers [11]. A number of factors other than dietary Zn concentration and stress conditions determine the amount of Zn needed for supplementation, including dietary phytate and Zn sources [5,12,13,14]. Sahin et al. [15] have extensively reviewed the role of zinc in alleviating the effect of heat stress in poultry. They demonstrated that zinc can improve the antioxidant status, immune response and nutrient digestibility of heat stressed chickens. Research reports indicated that the requirement for zinc is increased during exposure to heat stress conditions [15,16,17,18]. It is believed that zinc is essential in enzymes critical for the integrity of the cells involved in heat stress. Zinc oxide and zinc sulfate are two prevalent inorganic zinc sources for poultry feed supplementation. ZnO is highly stable, but less bioavailable for poultry than reagent-grade or feed-grade Zn sulfate [15]. The sulfates are highly water soluble, allowing reactive metal ions to promote free radical formation. This reaction can lead to the breakdown of vitamins and, ultimately, to the degradation of fats and oils, decreasing the nutrient value of the diet. Therefore, in most countries, using ZnO as an inorganic source of Zn is preferred by the industry. However, some researchers have reported that applying zinc oxide decreased nutrient digestibility, and that higher amounts of undigested nutrients in excreta is a threat for sustainable agriculture [19,20]. On the other hand, zinc sources should solubilize and change into ionic form in order to be absorbed by the enterocytes. However, ionic forms of zinc may be chelated by other substances and become unavailable for the birds [21]. Today, poultry nutritionists are searching for nutrients that have higher bioavailability to supply precision feeding, in order to reach optimum performance in the farm. Therefore, the present study was conducted to compare regular ZnO and the potentiated form of ZnO, HiZox^®^ (Animine), at different dosages on feed efficiency, growth performance, skin quality and bone quality of broilers suffering heat stress.

## 2. Materials and Methods

All of the methods applied in the present study were approved by Animal Science Department of Tehran University, and were in agreement with Iranian Council for Animal Care [22].

### 2.1. Birds

A total of one thousand two hundred male feather broilers (Ross 308) were provided by a commercial company (Derakhshan CO., Rasht, Iran) and used in a 42-day experiment. The density was 9.6 broilers per m^2^ (one bird/0.10416 m^2^). Chicks were randomly assigned to 40 experimental units.

### 2.2. Diets

Basal corn–soybean meal diets were formulated for starter (1–10 d), grower (11–24 d) and finisher (25–42 d) periods according to the Ross 308 nutritional guideline (Table 1, Table 2 and Table 3). Before formulating diets, the main ingredients were analyzed in order to evaluate their moisture, crude fat, crude protein, carbohydrate and ash content. Then, experimental diets were formulated by using WUFFDA software (https://secure.caes.uga.edu/extension/publications/files/html/RB438/WUFFDA-English.html accessed on 31 July 2022).

Basal diets supplemented with 75, 100 and 125 mg/kg zinc from HiZox (Animine, 335 Chemin du Noyer, 74330 Sillingy, France) and 100 mg/kg zinc from regular ZnO were used to make four treatments. In order to create a good mix, first zinc sources were mixed with 2 kg ground corn with 2 mm particle size, then added to a horizontal mixer to mix with other ingredients. The form of the diet was mash. The purity of the regular ZnO and HiZox was 76%, and their physical characteristics are shown in Table 4.

### 2.3. Samples Analysis

Protein (method no: 968/06), fat (method no: 948/16) and zinc content (method no: CAS-7440-66-6 zinc, 1995) of the diets was measured by following the procedure of AOAC. At the end of the experiment, blood samples were taken from ulnar vein of two birds per pen, and Zn content of plasma and alkaline phosphatase activity (ALP) was measured with an automatic biochemical analyzer (Hitachi 717, Boehringer Mannheim, Ingelheim am Rhein, Germany) using an Elitech Diagnostic kit (catalog no. A.110537). At slaughter, gizzard and jejunum contents were collected separately, and pH was measured by Testo 205 (Testo, Germany). After pH measurement, samples were lyophilized (Alpha 1-2 LD plus, Christ, Germany) and frozen (−20 °C). Samples of lyophilized digesta (0.3 g) were solubilized for 24 h at room temperature by using 10 mL 65% nitric acid, and then dried using a heater (300 °C). Before complete drying occurred, 2.5 mL of 72% perchloric acid (HClO_4_) was added, and dried material was acidified with 5 drops of 37% HCl in a 100 mL volume flask. Next, 50 mL distilled water was added, and the mixture was finally filtrated (Whatman NO 41). Filtrated material was diluted to 100 mL volume with H_2_O before Zn analysis. Total Zn was analyzed by an atomic absorption flame emission spectrophotometer (AA-670, Shimadzu, Japan). In order to determine the soluble Zn content in the digesta, lyophilized samples were rehydrated (0.3 g lyophilized digesta for 10 mL deionized water) and stirred constantly at 38 °C for 2 h. Supernatants were clarified two times by centrifugation (6000× *g*, 1 h, 20 °C) and separated by filtration (Whatman NO 41). The filtrated supernatant was acidified with one drop of HNO 16 N before Zn analysis. Soluble Zn was analyzed by atomic absorption flame emission spectrophotometer (AA-670, Shimadzu, Japan).

### 2.4. Experimental Design

A completely randomized design was used, with four treatments. Each treatment was replicated ten times, and each studied 30 birds. One-day-old broilers were randomly distributed into 40 floor pens. Each pen was equipped with 1 plastic pan feeder and 1 bell drinker and was covered by 5 cm wood shaving material. Birds received the mash diet from 1 to 42 days of age with free access to water and feed, and a 24-h photo schedule was applied. Weight gain, feed intake and livability of chicks were measured at days 7, 14, 28 and 42. At the end of the experimental period, two birds near the pen mean-body-weight were selected and, after slaughter, the carcass weight (including thighs, breast, abdominal fat and liver) was measured.

### 2.5. Induced Heat Stress

After the third week (21 d), in order to induce heat stress conditions in the broiler house, from 1 pm until 5 pm, the experimental house’s temperature was kept between 30–34 °C. The relative humidity was about 65 ± 5%. Panting behaviour of the birds was considered to be a sign of induced heat stress.

### 2.6. Skin Resistance Test

At 42 d of age, a total of 120 birds were slaughtered in the experimental slaughterhouse by using wet plucking, and an incision about 2 cm in length was made in the region between the thigh and the back. After defeathering, the incision was measured again. The difference in incision length before and after defeathering was recorded as skin tearing [24].

### 2.7. Footpad Dermatitis Score

Three types of footpad lesion were defined based on their severity. Type I were mild lesions, visually characterized by scale enlargement and erythema, and histologically by hyperplasia and hyperkeratosis of the epidermis, superficial dermal congestion and edema. Type II were moderate, superficial lesions, visually characterized by hypertrophic and hyperkeratotic scales covered with yellowish to brownish exudates, and histologically by a prominent pustular and crust-forming dermatitis. Type III lesions were the most pronounced, visually characterized by a thick dark adherent crust, and histologically by extensive ulceration [25].

### 2.8. Bone Physical Characteristics

After slaughter and carcass measurement, the left tibia of 80 birds were removed and adhered tissues were separated manually, then these were transferred to mechanic lab for further measurements. Length and diameter of tibia at the narrowest and widest points were measured by caliper (±0.01 mm).

According to beam theory, a three-point bending test was performed for measuring the bone physical characteristics following the ASABE standard procedure (Approved February 1993; reaffirmed January 2007 as an American National Standard, page 699). This Standard is designed for use in determining the mechanical properties of animal bones, such as the ultimate shear strength, ultimate bending strength, apparent modulus of elasticity, and fracture energy. The elongation, extension, force (N) and energy (MJ) needed at the peak and break point of male broiler chicken’s tibia were measured by Santam, MRT-5 instrument [26,27].

### 2.9. Statistical Analysis

Data were subjected to 1-way ANOVA using the GLM procedure of SAS [28]. The word “pen” refers to the experimental unit. Significant treatment effects were separated by Duncan’s multiple range tests square treatment, and means were compared if a significant F statistic (10% level of *p*) was detected by ANOVA.

## 3. Results and Discussion

The effects of experimental diets on the body weight of male broiler chickens are shown in Table 5. The body weight of the birds that received diets supplemented with 75, 100 and 125 mg/kg zinc from HiZox and 100 mg/kg zinc from ZnO showed no significant difference at 7 or 14 days of age. In addition, the body weights of heat-stressed birds fed diets containing different levels of HiZox or ZnO were not different at 28 or 42 days of age. The effect of dietary zinc on feed intake at 7, 14, 28 and 42 days showed no statistically significant difference (Table 5). Induced heat stress diminished body weight and feed intake by approximately 17 and 21%, respectively. There was no significant effect of dietary treatment on the feed conversion ratio during the first and second weeks of rearing (Table 5).

However, for the rest of experimental period, the effect of treatments on the feed conversion ratio was significant (*p* < 0.05). At 28 days, chickens that received 125 mg/kg Zn from Hizox showed better feed efficiency (*p* < 0.05). At 42 days, broilers that consumed 75 or 100 mg/kg Zn from Hizox showed a higher feed conversion ratio, but broilers that received 125 mg/kg Zn from Hizox showed no significant difference compared to the birds that consumed 100 mg/kg zinc from ZnO (*p* < 0.05). In agreement with the results of the present study, De Grande et al. [29] reported that there was no significant difference among the feed intake or bodyweight of the broilers which were fed different sources of zinc. Elsayed et al. [30] evaluated the effect of different zinc sources (inorganic vs. organic) in low crude protein diets on the performance of quail chicks. The researchers found that the feed intake and the feed conversion ratio of the birds were not different among different experimental groups. Zinc bioavailability depends on the amount of other minerals or antagonists present in the poultry feed [31,32]. The effect of zinc on birds’ performance relates to different parameters, such as zinc amount, zinc source, birds’ age, specious, genus, diet components, such as phytate levels, etc. [33,34]. If the bird’s requirement for zinc is not supplied, feed intake and feed efficiency will decrease, and this will have a negative effect on the growth of the birds [35]. Regarding the present study, it seems that the need of broilers to zinc for feed intake and body weight gain is supplied in all diets, and perhaps lower levels of dietary zinc should be added to the diet in order to observe significant differences in feed intake or body weight gain of broilers. The mortality rate of heat-stressed male broiler chickens that received different dosages of HiZox was 2.85% less than the regular ZnO group (*p* < 0.06, Figure 1).

According to the weekly mortality record, in all groups, 30% of the total mortality belongs to the first three weeks. The data show that 35% of total mortality occurred in the first week of induced heat stress (21–28 d), and 35% of total mortality took place during the rest of experimental period (29–42 d). Bartlett and Smith [36] reported that the level of zinc in the diet did not significantly influence broiler growth performance. However, their findings confirmed that both cellular and humoral immune responses were compromised in birds exposed to heat stress, demonstrating the effects of high environmental temperature. Poultry house temperature is a critical subject, so much so that ignoring this factor can cause heat or cold stress in birds, and thus reduce birds’ production efficiency by altering their bodies’ physiological process [37]. Heat stress decreases intestinal epithelium integrity, increases permeability of pathogens into body cells and decreases nutrient uptake by gut cells [38]. When birds are rearing under heat stress conditions, oxidative stress also occurs, a high amount of reactive oxygen specious is produced and cellular disturbance occurs [38]. Kidd et al. [39] indicated that zinc supplementation in broilers’ diets above the 40 ppm threshold recommended by the National Research Council [6] enhanced antibody production. On the other hand, zinc can protect cell membranes from oxidative damage caused by free radicals. Oteiza et al. [40] suggested that Zn increases the synthesis of metallothionein, which acts as a free radical scavenger. Liu et al. [41] reported that zinc supplementation increased gene expression of superoxide dismutase, which contains zinc in the liver and thighs of broilers, and this led to decreased malondialdehyde concentration in the liver and muscles. The present study showed that HiZox had higher potential to alleviate the mortality of broilers compared with ZnO, which can be due to the higher ability of HiZox to scavenge free radicals, so the detrimental effect of heat stress on broilers’ livability is suppressed. HiZox at different dosages and 100 mg/kg regular zinc oxide had no effect on carcass yield, breast, thighs or abdominal fat percentage of chickens under heat stress (Table 6). However, HiZox, in comparison to regular zinc oxide, increased liver weight (*p* < 0.05, Table 6). In contrast with our results, previous researchers showed that zinc supplementation in the form of zinc-sulphate or zinc-picolinate had a positive effect on the performance and carcass traits of quails suffering from heat stress conditions [42].

Qudsieh et al. [43] reported that carcass yield of broilers consumed zinc supplement was not different from the control group. However, the breast weight of male birds who consumed the 240 mg zinc/kg diet was higher than the breast weight of male birds who consumed the 120 mg zinc/kg diet. Parallel to the results of the present study, some researchers have reported there was no significant difference in carcass yield of the birds fed supplemental zinc [44,45,46]. The results showed that addition of HiZox to the diets of male broilers under heat stress doubled the skin resistance during feather plucking in the slaughter plant and improved carcass quality (*p* < 0.07, Figure 2).

Zinc is a mineral that plays an important role in skin and bone formation [47]. It improves collagen and keratin synthesis in cartilage and bones [48,49]. Rossi et al. [24] reported that an increase in collagen content of the skin was found in broilers fed higher levels of organic Zn, without any beneficial effect on bird performance. Collagen is the major constituent of skin. The higher collagen content in birds fed higher levels of organic Zn serve to hold the cells more tightly together [50]. Furthermore, collagen gives flexibility and resilience to the skin and improved carcass appearance in our study. A previous study showed that adding dietary zinc improved tibia and back skin dermis thickness [50]. Bartlett and Smith [36] found that adding zinc (34, 68 or 181 mg/kg diet) to broilers’ diets under heat stress did not influence plasma zinc concentration in a way that is similar to the present results. In agreement with this result, Sharideh et al. [51] found that zinc supplementation (ZnO; 30, 60, 90 and 120 mg/kg diet) had no effect on alkaline phosphatase activity in aged broiler breeders. They stated that perhaps the levels of zinc used were higher than what is needed to influence alkaline phosphatase activity. Today, footpad dermatitis score is considered to be a poultry welfare index in farms. Chen et al. [52] found that zinc, copper and manganese supplementation shows a positive effect on broilers’ performance and footpad injuries by improving collagen synthesis, deposition, organization, cell migration, matrix remodeling, angiogenesis and regulation of inflammation. However, zinc supplementation, in the present study, had no effect on footpad dermatitis score, and it seems that a lower amount of zinc should be studied in order to determine the level of its effectiveness on such a critical welfare index. Zinc intestinal absorption was significantly regulated via efflux and influx into intestinal cells by gene expression [53,54,55]. Jondreville et al. [13] reported that 59 mg Zn/kg maximized broiler plasma and bone Zn concentration. Huang et al. [56] indicated that the maximum broiler weight gain was observed in the diet supplemented with 20 mg of Zn/kg (48.37 mg/kg, total dietary Zn). Thus, supplementation of the diet with more than 75 mg/kg in the present study could not improve performance and plasma Zn, as the supply was already more than the requirement. Tibia weight, length and narrowest and widest diameter were not affected by supplemented zinc (Table 6). Tibia breaking strength, which included elongation and extension, improved with consumption of the diet supplemented with 75 mg/kg zinc from HiZox (*p* < 0.09, Table 7). The HiZox-75 broilers required higher amounts of energy (MJ) for tibia breaking at the break and peak point at 42 days (*p* < 0.09; *p* < 0.07, Figure 3).

Tibia breaking required more force (N) in HiZox-75 broilers, but this effect was not statistically significant (Table 7). Previous experiments have found that applying dietary zinc improved bone characteristics [57], which is due to its effect on protein synthesis [58,59], and zinc acts as a hormonal growth mediator, with effects such as influencing insulin-like growth factor I on osteoblasts [60]. Researchers have studied bone strength by assessing its reaction to different challenges. Tibia strength depends on several factors, such as diet components, age, sex, environmental conditions, etc. All mentioned parameters have an effect on the expression of collagen and proteins, content of the mineral material and structure of the bone [60]. Vakili and Rashidi [61] reported that adding fish oil, zinc and vitamin E to broilers’ diets under heat stress improved the tibia strength of the birds. Nguyen et al. [11] reported that zinc supplementation improved tibia strength, gene expression of claudin-1 and tight junction protein 1 as gut integrity biomarkers in broiler chickens. Zinc concentration in the broiler gizzard and jejunum were affected by dietary Zn content (Table 8). Jejunum Zn concentrations reflected the quantity of ingested Zn (*p* < 0.0001).

Zinc solubility has a direct effect on its absorption from epithelial cells; thus, the chemical form of zinc and diet components (including preventor or promotor agents) has a direct effect on zinc absorption [56]. Higher solubility of zinc increases its absorption from the intestine. Amino acids or chelators (e.g., EDTA) can bind to zinc and increase its solubility, thus increasing the zinc absorption [62]. Gizzard Zn solubility was dependent on dietary treatment (*p* < 0.03). Solubility of Zn in the gizzards of chickens who received Zn from HiZox near 30% was higher than that of chickens fed regular ZnO. Thus, the rate of HiZox absorption is higher that ZnO, and it can be helpful to supply zinc precision feeding and decrease Zn excretion into the environment.

## 4. Conclusions

It seems that Zn from HiZox can be more efficient in decreasing heat stress mortality and increasing skin resistance and bone breaking strength. Perhaps these observations are due to different levels of efficacy and patterns of solubility of zinc between HiZox and regular ZnO in the gastrointestinal tracts of broiler chickens. Further research is needed to clarify the effects of Hizox on the body’s biochemical and physiological processes.

## Figures and Tables

**Figure 1 animals-12-03272-f001:**
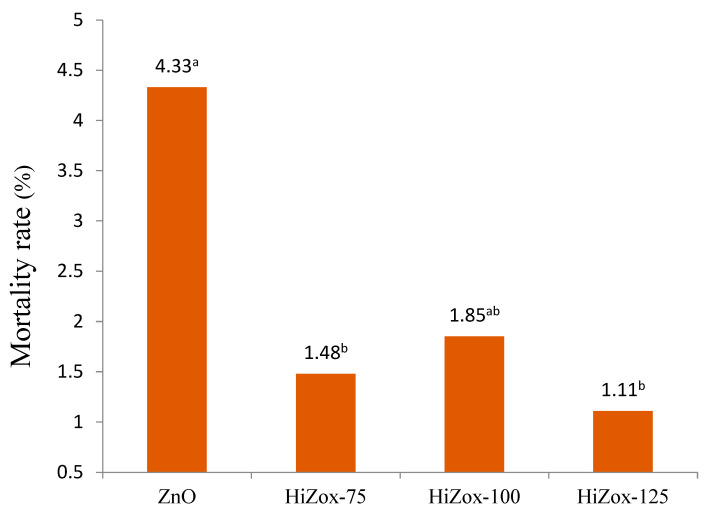
Effect of experimental diets on mortality rate of heat-stressed male broiler chickens between 1 and 42 days (*p* < 0.06, SE 0.06). ^a^, ^b^ Means in a row with different superscripts differ significantly (*p* < 0.05).

**Figure 2 animals-12-03272-f002:**
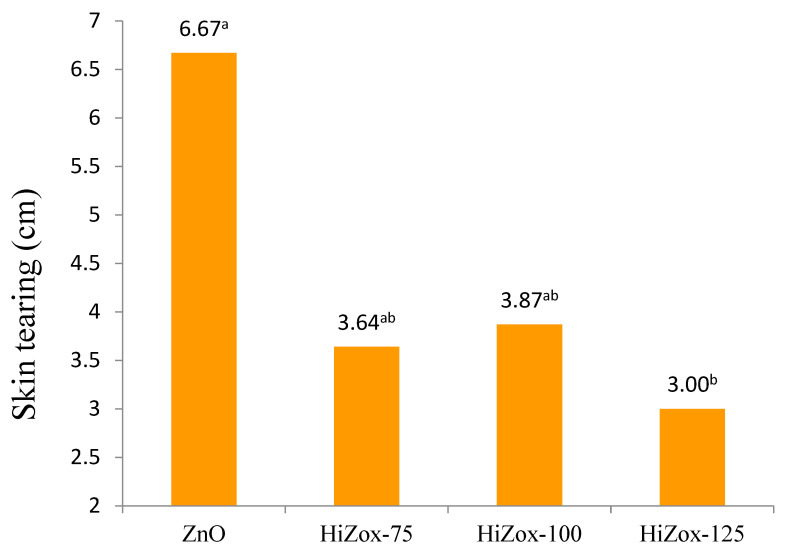
Effect of experimental diets on skin resistance test of heat stressed male broiler chicken at 42 day (*p* < 0.07, SE 1.01). ^a^, ^b^ Means in a row with different superscripts differ significantly (*p* < 0.05).

**Figure 3 animals-12-03272-f003:**
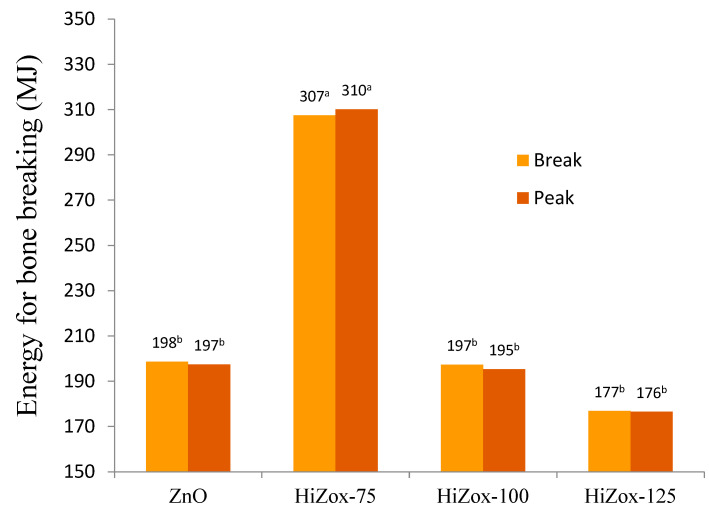
Effect of experimental diets on energy (MJ) required for bone breaking at break and peak points of heat-stressed male broiler chicken at 42 days (*p* < 0.09, SE 29.4; *p* < 0.07, SE 29.5). ^a^, ^b^ Means in a row with different superscripts differ significantly (*p* < 0.05).

**Table 1 animals-12-03272-t001:** Ingredients and nutrient content of experimental starter (1–10 days) diets ^1^.

Ingredients (g/kg)	Treatment			
	ZnO-100	HiZox-75	HiZox-100	HiZox-125
Corn grain	505.5	505.5	505.5	505.5
Soybean meal	408	408	408	408
Corn oil	39.9	39.9	39.9	39.9
Di calcium phosphate	20.9	20.9	20.9	20.9
Oyster shell	8.7	8.7	8.7	8.7
Common salt	3.7	3.7	3.7	3.7
Sodium bicarbonate	1.5	1.5	1.5	1.5
Vitamin supplement ^2^	2.5	2.5	2.5	2.5
Mineral supplement ^2^	2.5	2.5	2.5	2.5
D L Methionine	3.5	3.5	3.5	3.5
L Lysine HCl	2	2	2	2
L Threonine	1.3	1.3	1.3	1.3
Sum	1000	1000	1000	1000
ZnO ^3^	0.13158	-	-	-
HiZox ^3^	-	0.098680	0.13158	0.16447
Nutrients (calculated %)				
ME_n_ (kcal/kg)	3000	3000	3000	3000
Crude protein	23	23	23	23
Calcium	0.96	0.96	0.96	0.96
Available phosphorus	0.48	0.48	0.48	0.48
Na	0.2	0.2	0.2	0.2
(Na + K)-Cl (meq/kg)	250	250	250	250
Dig. Lys ^4^	1.28	1.28	1.28	1.28
Dig. Met	0.65	0.65	0.65	0.65
Dig. Met + Cys	0.95	0.95	0.95	0.95
Dig. Thr	0.86	0.86	0.86	0.86
Nutrients (analyzed %)				
Crude protein	23.12	23.50	22.07	23.39
Crude fat	6.1	6.5	6.7	6.4
Zinc (ppm) ^5^	126.3	113.45	137.40	183.06

^1^ Diets formulated on an as-fed basis and nutrient specification, followed by ROSS 308 (2014) management guides. ^2^ Vitamin and mineral premix provided the following per kilogram of diet: Vitamin A: 12,000 IU; Cholecalciferol: 5000 IU; Vitamin E:80 IU; Vitamin k3: 3.2 mg; Vitamin B12: 0.017 mg; Biotin: 0.22 mg; Folacin: 2.2 mg; Niacin: 65 mg; Pantothenic acid: 20 mg; Pyridoxine: 4.3 mg; Riboflavin: 8.6 mg; Thiamine: 3.2 mg. Choline (as choline chloride): 400 mg, Copper (as cupric sulfate 5H2O): 16 mg; Iodine (as calcium iodide): 1.25 mg; Iron (as ferrous sulfate 4H2O): 20 mg; Manganese (as manganese sulfate): 120 mg; Selenium (as sodium selenate): 0.3 mg, and no added Zinc. ^3^ ZnO (76%) and HiZox (76%) added to mineral supplement contained carrier. ^4^ Calculated amino acid composition is reported on a standardized ileal digestible amino acid basis (NIR spectroscopy). ^5^ The amount of analyzed Zn in the sample taken before mixing the source of zinc was 26.4 ppm.

**Table 2 animals-12-03272-t002:** Ingredients and nutrient content of experimental grower (11–24 days) diets ^1^.

Ingredients (g/kg)	Treatment			
	ZnO-100	HiZox-75	HiZox-100	HiZox-125
Corn grain	536.8	536.8	536.8	536.8
Soybean meal	372	372	372	372
Corn oil	49.5	49.5	49.5	49.5
Di calcium phosphate	18.7	18.7	18.7	18.7
Oyster shell	8	8	8	8
Common salt	2.4	2.4	2.4	2.4
Sodium bicarbonate	2.3	2.3	2.3	2.3
Vitamin supplement ^2^	2.5	2.5	2.5	2.5
Mineral supplement ^2^	2.5	2.5	2.5	2.5
D L Methionine	3	3	3	3
L Lysine HCl	1.4	1.4	1.4	1.4
L Threonine	0.9	0.9	0.9	0.9
Sum	1000	1000	1000	1000
ZnO ^3^	0.13158	-	-	-
HiZox ^3^	-	0.098680	0.13158	0.16447
Nutrients (calculated %)				
ME_n_ (kcal/kg)	3100	3100	3100	3100
Crude protein	21.5	21.5	21.5	21.5
Calcium	0.87	0.87	0.87	0.87
Available phosphorus	0.43	0.43	0.43	0.43
Na	0.17	0.17	0.17	0.17
(Na + K)-Cl (meq/kg)	247	247	247	247
Dig. Lys ^4^	1.15	1.15	1.15	1.15
Dig. Met	0.59	0.59	0.59	0.59
Dig. Met + Cys	0.87	0.87	0.87	0.87
Dig. Thr	0.77	0.77	0.77	0.77
Nutrients (analyzed %)				
Crude protein	21.09	21.04	21.4	21.90
Crude fat	7.6	7.2	7.1	7.4
Zinc (ppm) ^5^	120.95	98.95	128.15	158.35

^1^ Diets formulated on an as-fed basis and nutrient specification, followed by ROSS 308 (2014) management guides. ^2^ Vitamin and mineral premix provided the following per kilogram of diet: Vitamin A: 12,000 IU; Cholecalciferol: 5000 IU; Vitamin E:80 IU; Vitamin k3: 3.2 mg; Vitamin B12: 0.017 mg; Biotin: 0.22 mg; Folacin: 2.2 mg; Niacin: 65 mg; Pantothenic acid: 20 mg; Pyridoxine: 4.3 mg; Riboflavin: 8.6 mg; Thiamine: 3.2 mg. Choline (as choline chloride): 400 mg, Copper (as cupric sulfate 5H2O): 16 mg; Iodine (as calcium iodide): 1.25 mg; Iron (as ferrous sulfate 4H2O): 20 mg; Manganese (as manganese sulfate): 120 mg; Selenium (as sodium selenate): 0.3 mg, and no added Zinc. ^3^ ZnO (76%) and HiZox (76%) added to mineral supplement contained carrier. ^4^ Calculated amino acid composition is reported on a standardized ileal digestible amino acid basis (NIR spectroscopy). ^5^ The amount of analyzed Zn in the sample taken before mixing the source of zinc was 26.4 ppm.

**Table 3 animals-12-03272-t003:** Ingredients and nutrient content of experimental finisher (25–42 days) diets ^1^.

Ingredients (g/kg)	Treatment			
	ZnO-100	HiZox-75	HiZox-100	HiZox-125
Corn grain	585.4	585.4	585.4	585.4
Soybean meal	320	320	320	320
Corn oil	56.3	56.3	56.3	56.3
Di calcium phosphate	16.8	16.8	16.8	16.8
Oyster shell	7.4	7.4	7.4	7.4
Common salt	2.3	2.3	2.3	2.3
Sodium bicarbonate	2	2	2	2
Vitamin supplement ^2^	2.5	2.5	2.5	2.5
Mineral supplement ^2^	2.5	2.5	2.5	2.5
D L Methionine	2.7	2.7	2.7	2.7
L Lysine HCl	1.4	1.4	1.4	1.4
L Threonine	0.7	0.7	0.7	0.7
Sum	1000	1000	1000	1000
ZnO ^3^	0.13158	-	-	-
HiZox ^3^	-	0.098680	0.13158	0.16447
Nutrients (calculated %)				
ME_n_ (kcal/kg)	3200	3200	3200	3200
Crude protein	19.5	19.5	19.5	19.5
Calcium	0.79	0.79	0.79	0.79
Available phosphorus	0.39	0.39	0.39	0.39
Na	0.16	0.16	0.16	0.16
(Na + K)-Cl (meq/kg)	221	221	221	221
Dig. Lys ^4^	1.03	1.03	1.03	1.03
Dig. Met	0.54	0.54	0.54	0.54
Dig. Met + Cys	0.80	0.80	0.80	0.80
Dig. Thr	0.69	0.69	0.69	0.69
Nutrients (analyzed %)				
Crude protein	19.20	19.80	19.61	19.25
Crude fat	8.1	8.0	8.3	8.1
Zinc (ppm) ^5^	119.85	97.15	129.01	151.55

^1^ Diets formulated on an as-fed basis and nutrient specification followed by ROSS 308 (2014) management guides. ^2^ Vitamin and mineral premix provided the following per kilogram of diet: Vitamin A: 12,000 IU; Cholecalciferol: 5000 IU; Vitamin E:80 IU; Vitamin k3: 3.2 mg; Vitamin B12: 0.017 mg; Biotin: 0.22 mg; Folacin: 2.2 mg; Niacin: 65 mg; Pantothenic acid: 20 mg; Pyridoxine: 4.3 mg; Riboflavin: 8.6 mg; Thiamine: 3.2 mg. Choline (as choline chloride): 400 mg, Copper (as cupric sulfate 5H2O): 16 mg; Iodine (as calcium iodide): 1.25 mg; Iron (as ferrous sulfate 4H2O): 20 mg; Manganese (as manganese sulfate): 120 mg; Selenium (as sodium selenate): 0.3 mg, and no added Zinc. ^3^ ZnO (76%) and HiZox (76%) added to mineral supplement contained carrier. ^4^ Calculated amino acid composition is reported on a standardized ileal digestible amino acid basis (NIR spectroscopy). ^5^ The amount of analyzed Zn in the sample taken before mixing the source of zinc was 23.5 ppm.

**Table 4 animals-12-03272-t004:** Physical characteristics of the regular ZnO and HiZox (Adapeted with permission from Noori et al. [23]. Copyright 2019, Noori).

Characteristics	Regular ZnO	Hizox
Particle size (nm)	100–1000	<100
Area to weight ratio (m^2^/g)	2.4	42
CV ^1^ (%)	5.61	3.65
Angle of repose (degree)	35	28
Mixability	poor	good

^1^ Coefficient of variation of Zn in complete feed.

**Table 5 animals-12-03272-t005:** Effect of experimental diets on live body weight, daily feed intake and feed conversion ratio of male broiler chickens (g).

	ZnO-100	HiZox-75	HiZox-100	HiZox-125	SEM	*p*-Value
1–7 days						
Live body weight (g)	159.9	160.8	159.2	158.1	1.9	0.8
Average daily feed intake (g)	19.9	19.7	19.9	20.0	0.2	0.87
Feed conversion ratio	0.87	0.86	0.88	0.89	0.01	0.26
1–14 days						
Live body weight (g)	422.1	420.4	420.6	417.1	5.1	0.71
Average daily feed intake (g)	33.7	33.7	33.7	32.7	0.3	0.29
Feed conversion ratio	1.11	1.11	1.11	1.10	0.01	0.9
1–28 days						
Live body weight (g)	1355.4	1378.0	1342.7	1364.4	20.3	0.66
Average daily feed intake (g)	65.5	65.2	64.6	64.1	0.8	0.63
Feed conversion ratio	1.34 ^a^	1.32 ^ab^	1.34 ^a^	1.31 ^b^	0.007	0.01
1–42 days						
Live body weight (g)	2504.5	2509.3	2471.0	2461.6	24.1	0.41
Average daily feed intake (g)	91.7	91.5	90.5	90.2	1.08	0.72
Feed conversion ratio	1.52 ^b^	1.55 ^a^	1.55 ^a^	1.54 ^ab^	0.007	0.04

^a^, ^b^ Means in a row with different superscripts differ significantly (*p* < 0.05).

**Table 6 animals-12-03272-t006:** Effect of experimental diets on carcass fractional weights (% of live weight), plasma zinc content, Alkaline Phosphatase (ALP), footpad dermatitis score, acidity of the gizzard and small intestine and physical characteristics of the tibia of male broiler chickens (g).

	ZnO-100	HiZox-75	HiZox-100	HiZox-125	SEM	*p*-Value
Carcass fractional weights						
Carcass	75.6	75.6	75.4	75.5	0.28	0.95
Breast	26.8	27.2	26.8	26.8	0.38	0.84
Thighs	21.0	21.0	20.6	20.7	0.17	0.20
Abdominal fat pad	1.16	1.21	1.27	1.33	0.08	0.45
liver	1.67 ^b^	1.80 ^ab^	1.90 ^a^	1.83 ^ab^	0.05	0.04
Plasma Zn (µg/dL)	103.1	103.8	100.7	106.0	2.4	0.55
ALP (u/L)	5282	5356	4619	5448	392	0.44
Footpad dermatitis score	1.19	1.40	1.55	1.48	1.5	0.30
Gizzard pH	2.59	2.56	2.65	2.61	0.08	0.90
Small intestine pH	5.92	5.95	5.87	6.01	0.13	0.89
Tibia weight (g)	6.99	7.17	6.96	6.84	0.16	0.56
Tibia length (mm)	98.2	99.3	98.6	97.5	0.09	0.53
Tibia diameter (widest, mm)	9.0	9.5	9.3	9.3	0.01	0.36
Tibia diameter (narrowest, mm)	7.4	7.6	7.7	7.5	0.01	0.54

^a^, ^b^ Means in a row with different superscripts differ significantly (*p* < 0.05).

**Table 7 animals-12-03272-t007:** Effect of experimental diets on elongation, force and extension of the tibia in male broiler chickens at peak and break point (d 42).

Treatment	Peak			Break		
	Elongation (%)	Force (N)	Extension (mm)	Elongation (%)	Force (N)	Extension (mm)
ZnO-100	1.0381 ^b^	247.83	1.5571 ^b^	1.0623 ^b^	247.66	1.5934 ^b^
HiZox-75	1.4517 ^a^	278.72	2.1776 ^a^	1.4649 ^a^	273.37	2.1974 ^a^
HiZox-100	1.0325 ^b^	243.31	1.5488 ^b^	1.0344 ^b^	242.67	1.5516 ^b^
HiZox-125	1.0000 ^b^	229.47	1.5000 ^b^	1.0034 ^b^	229.12	1.5051 ^b^
SEM	0.11	14.7	0.16	0.11	14.9	0.16
*p*-Value	0.10	0.30	0.10	0.09	0.40	0.09

^a^, ^b^ Means in a column with different superscripts differ significantly (*p* < 0.05).

**Table 8 animals-12-03272-t008:** Effect of experimental diets on Zn status of digesta in male broiler chickens (d 42).

Treatment	Gizzard			Jejunum		
	Zn (mg/kg DM)	Soluble Zn (mg/kg DM)	Zn Solubility (%)	Zn (mg/kg DM)	Soluble Zn (mg/kg DM)	Zn Solubility (%)
ZnO-100	64.1 ^a^	17.5	29.1 ^b^	291.6 ^b^	35.1	12.2
HiZox-75	53.6 ^ab^	19.1	37.7 ^ab^	222.1 ^c^	25.3	12.5
HiZox-100	41.3 ^b^	19.1	45.7 ^a^	387.3 ^a^	33.1	8.9
HiZox-125	64.5 ^a^	17.3	33.0 ^b^	420.9 ^a^	37.4	9.4
SEM	6.7	1.5	4.3	24.3	3.2	1.2
*p*-Value	0.08	0.78	0.03	0.0001	0.09	0.11

DM = dry matter. Calculated as the ratio of total to soluble Zn concentration. ^a^, ^b^, ^c^ Means in a column with different superscripts differ significantly (*p* < 0.05).

## Data Availability

Not applicable.

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
