# Peer review of "Evaluation of an Innovative Zn Source on Feed Efficiency, Growth Performance, Skin and Bone Quality of Broilers Suffering Heat Stress"

_animals, 2022, doi:10.3390/ani12233272_

Round 1

Reviewer 1 Report

This study investigates different doses of a commercial Zinc product (Hizox) as compared with only one level (100 mg/kg) supplementation of conventional ZnO. I appreciate the work done with a large number of broilers and though that it may contribute to practice in the area.

However, the manuscript has several shortcomings that makes controversial the acceptability of it.  Novel information produced from the study is quite limited. Overall mortality seems interesting but distribution of mortality among the treatments are not presented.   

Other concerns are summarized in below.  

 The study design and statistics do not allow us to see clearly the effect of different doses of commercial potentiated ZnO.  Only one levels of conventional ZnO was compared with different doses of a commercial product (Hizox). However, it could be possible to have a clear information regarding the doses of Hizox if a polynomial regression analysis would have been conducted. Beside, main conclusions from the study rely on the edge of 0.10 probability (skin quality p<0.07, mortality p<0.06, bone quality p<0.09) and seems to be quite ambitious.

M&M section: It needs more detailed description of heat stress conditions induced and slaughter conditions regarding de-feathering which is   definitely affects skin tearing.  Experimental or commercial slaughterhouse? Dry or wet plucking?  etc.  

Results: First, it has been concluded that total mortality (0-42 d) decreased with all levels of HiZox supplementation as compared to regular ZnO of 100 mg/kg under heat stress conditions.  Did mortality data from different periods of growth (0-21, 21-28, 21-42 etc.) have the similar trend? These data should be presented in the paper to make a sound conclusion.

Second, heat stress was commenced on d 21. Therefore, presentation of BW and other performance traits on d 21 and related periods are crucial. BW, feed consumption, and FCR before and just after heat stress commenced would be more informative.    

FCR data are confusing. On d 28, the best FCR was observed in HiZox -125 group but at the end of the experiment the best FCR was observed in regular ZnO. It would be a more satisfactory discussion if data between 21-28 d have been presented. And discussion on lines 216-222, 241-244 are not sound enough.  

 Minor issues:

Line 16 : ..heat stress birds please change to ..heat stressed birds

Line 51: please correct …reaches to research

Line 51-53 and Lines 63-66 should be given together for accumulating information regarding the importance of Zn under heat stress. 

Line 141, Line 175: Figure 1 and Figure 2 should be moved to results and discussion section and properly cited.

Lines 193-196: Contradictory to the information given in M&M section on line 102, in which it was stated that diets were in mash form. 

Line 285-287: The sentence needs to be rephrased in order to clarification.

Line 297-298: No need to discuss insignificant data.

Line 316: As it mentioned about traits different from bone parameters, a new paragraph is necessary.

Line 320: More Zn was ingested in jejunum with higher levels of HiZox. But I think it is more worthy to present and discuss on data together regarding economics and environmental impact of the supplemented diets in the experiment.    

Author Response

Thank you for your valuable comments. We tried to do our best by editing the manuscript regards to the valuable comments.

Best Regards,

Reviewer 2 Report

The paper is evaluating an innovative Zn source on feed efficiency, growth performance, skin and bone quality of broilers suffering heat stress. This is an interesting topic but the manuscript can not be accepted in its current form and needs corrections and justification to be suitable for publication in ANIMLS. My major concerns are:

1- The manuscript needs intensive English language editing.

2- The authors used 1200 birds in this experiment which is sufficient however, during sampling they depended on A NON-REPRESENTITIVE sample. 80 birds is not enough to judge on such commercial product. 

3- The authors provided only the physical characterization of  HiZox so, chemical analysis of the HiZox  must be provided too. 

4- The authors depended on p< 0.1 as a significant level which is not common. 

5- The HiZox did not alter the bird's body weight although its effect on feed conversion ratio, The authors explained this by different zinc bioavailability depending on the amount of other minerals or antagonists present in the poultry feed. Thus, the chemical analysis of HiZox must be provided. 

6- in L 214-218, the authors mentioned that there were no effect of HiZox supplementation on body weight to the low level of Zn provided. However, they recommended to decrease the level of supplemented Zn! This is confusing and weird. 

7- in L 241-243, the authors emphasized that the lower mortality  level of HiZox was due to the higher ability of HiZox to scavenge free radicals??? How did they prove this?? Need confirmation. For examples measuring level of some antioxidant enzymes and malondyhyde contents. 

8- Figure 2 is quite similar to that found in this link 

https://animine.eu/wp-content/uploads/2019/07/1806-Poultry-production-Focus-Asia.pdf

9- The economic impact must be evaluated. 

10- The authors must provide a Conflict of interest. 

11- in conclusion, they strictly confirmed the improving effect of HiZox despite the non significant differences of most of the results. 

Author Response

(The authors gave the same response as above.)

Round 2

Reviewer 1 Report

Following issues need to be considered: 

1. The main conclusions from the study rely on the edge of 0.10 probability (skin quality p<0.07, mortality p<0.06, bone quality p<0.09). 

2. Revised version of the MS does not answer the questions regarding detailed description of heat stress conditions induced and slaughter conditions regarding de-feathering which is   definitely affects skin tearing.  Dry or wet plucking?

3. My comments on the presentation of data were not considered by the authors without any reasoning.  Heat stress was commenced on d 21. Therefore, presentation of BW and other performance traits, before and just after heat stress commenced, would be more informative. Thus, a more satisfactory discussion would be  if 21-28 d data have been presented.

Author Response

-Dear reviewer 1, thank you so much for your kind attention. We tried our best to do your valuable comments.

-Results of the study rely on P<0.1.

-M&M section: After third week (21 d), in order to induce heat stress condition in broiler house, from 1 pm until 5 pm experimental house’s temperature kept between 30-34â—¦C.

-Experimental slaughterhouse by using wet plucking was used.      

Results: First, it has been concluded that total mortality (0-42 d) decreased with all levels of HiZox supplementation as compared to regular ZnO of 100 mg/kg under heat stress conditions.  Did mortality data from different periods of growth (0-21, 21-28, 21-42 etc.) have the similar trend?

Yes it has similar trend.

Second, heat stress was commenced on d 21. Therefore, presentation of BW and other performance traits on d 21 and related periods are crucial. BW, feed consumption, and FCR before and just after heat stress commenced would be more informative. 

Unfortunately, we don’t have further data.   

 Minor issues:

Line 16: heat stress birds changed to heat stressed birds

Line 51: reaches changed to research

Line 51-53 and Lines 63-66 are given together for accumulating information regarding the importance of Zn under heat stress. 

Line 141, Line 175: Figure 1 and Figure 2 moved to results and discussion section and properly cited.

Lines 193-196: It is corrected and diets were in mash form. 

Line 285-287: The sentence is rephrased in order to clarification.

Line 297-298: Discussion of insignificant data removed.

Line 316: A new paragraph is considered.

Reviewer 2 Report

Dear authors, 

It seems that you did not consider any of reviewers' comments. 

Please provide your reply. 

regards

Author Response

Dear reviewer 2, thank you so much for the valuable comments. We tried our best to do them.

2- We use 120 birds to evaluate the effects of the treatments.

3- The authors provided only the physical characterization of  HiZox so, chemical analysis of the HiZox  must be provided too.  Animine company did not publish HiZox chemical analysis.

4- We considered data significant as P<0.1. 

6- in L 214-218, the conclusion is corrected. 

7- in L 241-243, the authors emphasized that the lower mortality level of HiZox was due to the higher ability of HiZox to scavenge free radicals??? How did they prove this?? Need confirmation. For examples measuring level of some antioxidant enzymes and malondyhyde contents. 

This manuscript is derived from our MSc student thesis, Mr. Mehrvarz. We have conducted another two projects on Hizox and we have seen its effect on reducing MDA level of the serum, however still we did not publish it in any journal.

10- Conflict of interest is added.

11- We corrected conclusion section.

Round 3

Reviewer 2 Report

I recommend this manuscript for publication in Animals.